# Efficient Poisson's Ratio Evaluation of Weft-Knitted Auxetic Metamaterials

Kun Luan [1], Zoe Newman [2], Andre West [2,*], Kuan-Lin Lee [3] and Srujan Rokkam [3]

1   Department of Textile Engineering, Chemistry and Science, Wilson College of Textiles, NC State University, Raleigh, NC 27606, USA; kluan@ncsu.edu
2   Department of Textile and Apparel, Technology and Management, Wilson College of Textiles, NC State University, Raleigh, NC 27606, USA; zlnewman@alumni.ncsu.edu
3   Advanced Cooling Technologies, Inc., Lancaster, PA 17601, USA; kuan-lin.lee@1-act.com (K.-L.L.); srujan.rokkam@1-act.com (S.R.)
*   Correspondence: ajwest2@ncsu.edu

**Abstract:** Auxetic metamaterials expand transversely when stretched longitudinally or contract transversely when compressed, resulting in a negative Poisson's ratio (NPR). Auxetic fabrics are 3D textile metamaterials possessing a unique geometry that can generate an auxetic response with respect to tension. In weft-knitted auxetic fabrics, the NPR property is achieved due to the inherent curling effect of the face and back stitches of the knit loops; they contract in an organized knitting pattern. The traditional method used to evaluate NPR is to measure the lateral fabric deformation during axial tensile testing on a mechanical testing machine, which is time-consuming and inaccurate in measuring uneven deformations. In this study, an efficient method was developed to evaluate the NPR of weft-knitted fabric that can also estimate deformation directionality. The elasticity and extension properties of the weft-knitted fabric can be analyzed immediately following removal from the knitting bed. Five fabrics, all with the same stitch densities (including four auxetic patterns and one single jersey pattern), were designed and produced to validate the proposed method. The use of our estimation method to evaluate the Poisson's ratio of such fabrics showed higher values compared with the traditional method. In conclusion, the deformation directionality, elasticity, and extensionality were examined. It is anticipated that the proposed method could assist in the innovative development and deployment of auxetic knitted metamaterials.

**Keywords:** auxetic fabric; weft knitting; metamaterials; negative Poisson's ratio; fabric deformation

## 1. Introduction

Negative Poisson's ratio (NPR) is a unique property rarely present in nature. Materials/structures with an NPR (auxetic metamaterials) exhibit extension (contraction) behavior perpendicular to uniaxial tension (compression) [1]. They are expected to demonstrate extraordinary extensionality and high energy absorption due to their low density, which allows the solid hinge-like elements to flex. These features found in NPR (auxetic) materials can be successfully utilized in biomedical devices [2], protective body armors [3], high-performance composites [4], sportswear [5], and packing material [6], among other uses.

Due to their organized interlaced pattern, textile materials naturally possess rigid and flexible characteristics, enabling them to be strong while retaining softness. As such, textiles are one of the preferred artificial structures suitable for producing auxetic materials. Metamaterials can be found in the form of filament bundles, yarn, fabric, or fiber-based composites [7]. Among those textiles, weft-knitted auxetic fabric is one of the most accessible and sustainable textile structures to design and manufacture using conventional yarns. Alternating the face and back loops in a repeated pattern (unit cell) provides contractions of the fabric to form the self-folded structure, which is generated by the unbalanced force of the knitting loop structure (the origin of the fabric curling) [1]. Interactions between

elastic elements and inelastic elements in a pattern with the NPR can shape the structure that is deformed in the direction perpendicular to the loading direction, in contrast with positive Poisson's ratio materials [8]. Various design patterns [9] have been used to produce weft-knitted auxetic fabrics with enhanced characteristics. Research into NPR textiles with regard to both structure and property has gained attention from textile engineers and designers, especially smart textiles researchers focusing on stimulated textile-based sensors or actuators [10].

Poisson's ratio is one of the elastic constants of materials in solid mechanics [11]. For anisotropic materials, such as conventional weft-knitted fabric [12], the Poisson's ratio in the wale/course direction represents the portion of the transverse strain to the axial strain. The methods for measuring Poisson's ratio include a mechanical testing method for solid materials (some of which can also be applied to soft materials) [13] and the unit strain method for soft materials. The mechanical test method can monitor the orthogonal deformations of solid/soft samples during tensile/compressive tests. The mechanical test can record axial deformation using a load sensor. To record lateral deformation, an additional monitoring device must be installed on the testing machine, such as an extensometer [14], a high-resolution camera (digital image correlation, DIC) [15], or a laser diffractographic device [16].

When applying tension to conventional/NPR knitting fabric during the tensile process, unlike a solid specimen, it will behave by narrowing/expanding along with stretching. The fabric deformation in the lateral direction gradually narrows or expands, reaching a critical value at the middle line of the fabric specimen [17]. The strain at the middle line can be divided by the axial strain to calculate the Poisson's ratio. Thus, the strain grid method needs to stretch the NPR fabric into a certain strain manually and measure the perpendicular strain [12]. The process may result in out-of-plane deformation and uneven stretching that may cause an underestimation of the fabric's NPR property [12]. The use of the mechanical testing method for obtaining the Poisson's ratio is time-consuming and requires extra strain measurement equipment to work along with the mechanical testing process. Textile designers and manufacturers are unlikely to spend increased effort on the development of NPR fabric because the NPR property requires additional characterizations. For both methods, the strain at the middle line is not accurately measured since the fixture of the clamps/hands limits the deformation on both ends of the fabric specimen. Furthermore, since the technology for producing auxetic materials is new in the "smart" textiles market, these issues significantly restrict the wide development and deployment of NPR fabrics.

The aim of this paper is to introduce a series of criteria housed in a method that can efficiently evaluate the NPR property of weft-knitted auxetic fabric. Dimensions in the wale direction and course direction of four auxetic fabrics and one plain knitted fabric were measured to calculate the NPR-related coefficients ($K_1$ and $K_2$, see definitions in Section 3), which were analyzed to estimate the NPR property and directionality of those fabrics. Only the knitting direction NPR was examined because it has a larger NPR compared with other directions. Furthermore, we used a traditional grid method to verify and validate the accuracy of the developed criteria. It is anticipated that when "smart" textiles flourish, our research on the rapid estimation of auxetic properties could facilitate the development of unique auxetic textile structures that exhibit tremendous potential to be implemented in the textile-based sensor, actuator, generator, and biomedical device industries.

## 2. Materials and Methods

### 2.1. Concept Description

Single jersey fabric is composed of knit loops intermeshing with one another, which is a widely deployed knitting structure in end-use productions in the textile industry [18]. It will shrink or curl after production [19] (once removed from the knitting machine) due to unbalanced forces generated by the curved yarn loop. By alternating loop faces, we can design and produce a zig-zag weft-knitted structure (auxetic fabric) that possesses an NPR [1].

Figure 1 illustrates the take-off shrinkage of single jersey fabric and the structural shrinkage of auxetic fabric. Take-off (doffed) shrinkage is a side effect of the unbalanced resilience force in single jersey fabric, often causing curling. To eliminate curling in regular products, a balanced structure, such as a rib-knit cuff, is normally employed to finish the edge of single jersey fabric. However, in the auxetic structure design, the resilient force of the knitting loop is redistributed by alternating knit-loop orientations. This generates an inherently balanced internal force, with the curling eliminated by the nature of the auxetic design. Auxetic fabric exhibits more shrinkage than single jersey fabric after being doffed from a knitting machine. The structure provides extra thickness for deformation when releasing tension on the fabric edges.

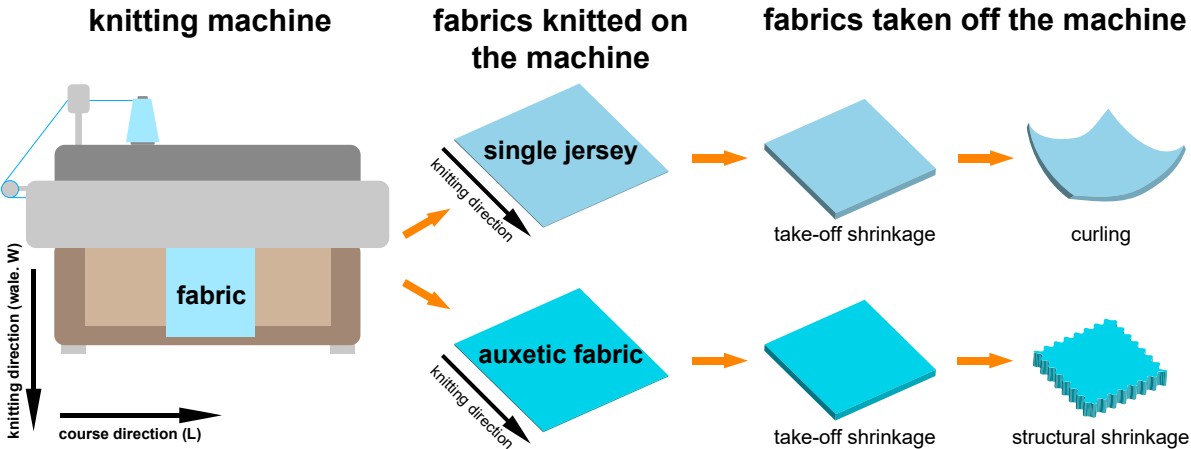

**Figure 1.** The take-off shrinkage of single jersey fabric versus the structural shrinkage of auxetic fabric.

To quantitively investigate the dimensional changes in auxetic fabrics, width (W), length (L), and thickness (T) were measured for NPR calculation and criteria validation.

Figure 2A indicates the dimensional change in the in-plane direction. The width and length of single jersey fabric can be measured after slightly stretching both ends. In contrast, auxetic fabric has an unmeasurable take-off shrinkage because it possesses an additional structural deformation in the thickness direction. In addition, the shrinkage of auxetic fabric results in difficulty in the distinguishment between take-off shrinkage and structural shrinkage. The NPR of auxetic fabric is difficult to obtain since it is caused by structural shrinkage, thereby limiting the accurate measurement of the mechanical property of auxetic fabric.

Figure 2B shows the side view of dimensional change in the thickness direction for single jersey fabric and auxetic fabric. Single jersey fabric shrinks following removal from the knitting machine due to the contraction of the knit loops in the in-plane direction, resulting in an increased thickness (T). Auxetic fabrics also exhibit increased thickness and have additional considerations due to structural contraction. Auxetic fabrics are thicker than single jersey fabrics, even if both are knitted under the same production conditions.

### 2.2. Negative Poisson's Ratio Criteria

Auxetic fabrics possess two dimensional changes in the in-plane direction and thickness measurements, including doffed shrinkage and structural shrinkage. The dimensional changes associated with doffed shrinkage are difficult to measure using the NPR calculation. To address this issue, a blanket was designed and produced that consisted of a single jersey structure and an auxetic pattern with the same stitch densities, as presented in Figure 3.

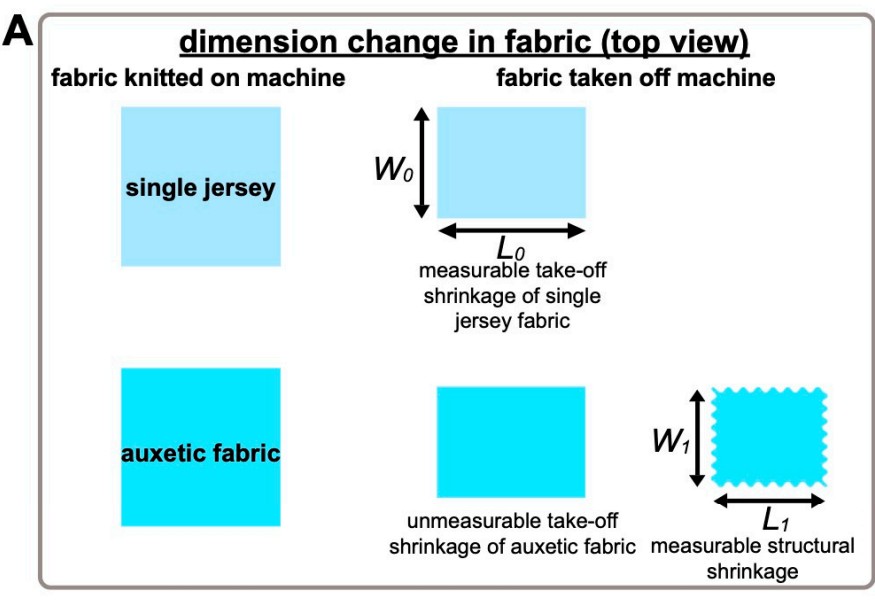

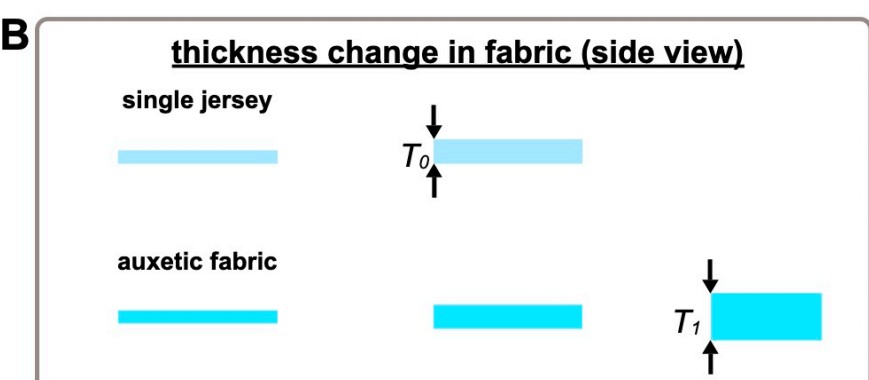

**Figure 2.** Dimensional change in single jersey fabrics and auxetic fabrics: (**A**) dimensional change in in-plane direction; (**B**) dimensional change in thickness.

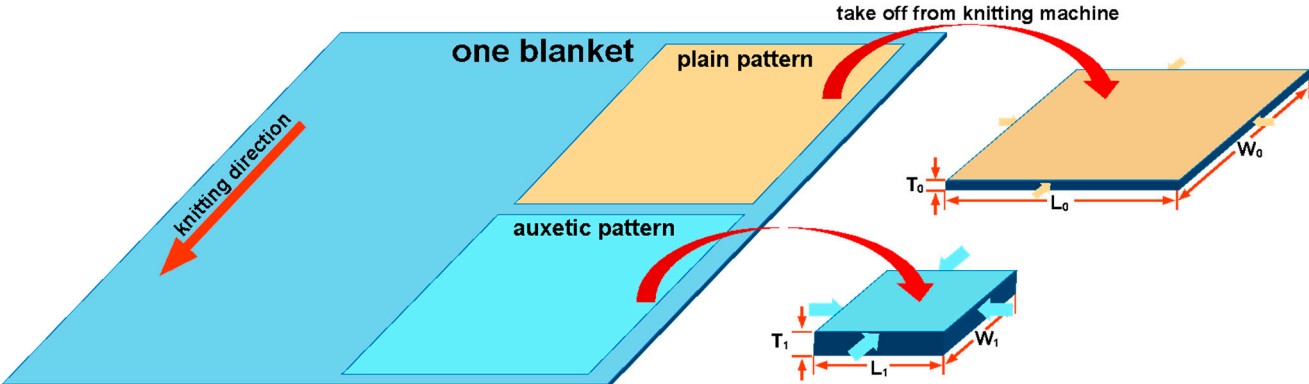

**Figure 3.** Design of the blanket that can efficiently estimate the NPR of the auxetic fabric.

Previous studies have proven that single jersey fabric is a positive Poisson's ratio material [20], meaning that the take-off shrinkage of single jersey cannot initiate NPR behavior. Since auxetic fabric possesses an NPR, its source of deformation is attributed to structural shrinkage. When stretching auxetic fabric in a planar orientation, the Poisson's ratio will become positive, the same as for single jersey fabric. Thus, through a comparison of the dimensional changes between single jersey fabric and auxetic fabric, we can calculate the Poisson's ratio of auxetic fabric using the following equations.

The definition of the Poisson's ratio in solid mechanics is

$$v_{12} = -\frac{\varepsilon_2}{\varepsilon_1} \tag{1}$$

Strain in the wale direction (knitting direction in Figure 1):

$$\varepsilon_2 = \frac{W_1 - W_0}{W_0} \tag{2}$$

Strain in the course direction (perpendicular to knitting direction):

$$\varepsilon_1 = \frac{L_1 - L_0}{L_0} \tag{3}$$

Therefore, the Poisson's ratio of auxetic fabric in Figure 3 in the in-plane direction can be obtained by

$$v_{12} = -\frac{L_0 \times (W_1 - W_0)}{W_0 \times (L_1 - L_0)} \tag{4}$$

where $W_0$ is the width of the single jersey fabric, $L_0$ is the length of the single jersey fabric, $W_1$ is the width of the auxetic fabric, and $L_1$ is the length of the auxetic fabric.

To examine the auxetic fabric directionality, we defined three dimensional coefficients:

$$K_1 = \frac{T_1 - T_0}{L_1 - L_0} \tag{5}$$

$$K_2 = \frac{W_1 - W_0}{L_1 - L_0} \tag{6}$$

$$K_3 = \frac{T_1 - T_0}{W_1 - W_0} \tag{7}$$

where $T_1$ is the thickness of the auxetic fabric and $T_0$ is the thickness of the single jersey fabric.

The physical meaning of $K_1$ is the ratio of thickness to length in the fabric length direction (course direction), in which a higher value means more shrinkage in length direction contributing to thickness.

$K_2$ is the ratio of dimensional change in the fabric width direction (wale direction) and fabric length direction (course direction). A higher $K_2$ value means a larger auxetic fabric aspect ratio. This may be used as a criterion to determine the usage of highly oriented auxetic fabrics in specific applications, such as an artificial blood vessel composed of weft-knitted auxetic fabric that requires anisotropic deformation during the pumping of fluid/solution.

$K_3$ is the ratio of thickness to length in the fabric width direction (wale direction), in which a higher value means more shrinkage in the width direction, contributing to thickness.

### 2.3. Fabric Design, Production, and Measurement

In the development of the auxetic structures, face and back loops and tucks were integrated into the fabric [1]. The alternate arrangement of the face and back loops results in a multidirectional shrinking phenomenon, creating the auxetic structure with a characteristic negative Poisson's ratio.

The SDS-ONE APEX3 Shima Seiki system version R-19 was used for the design and development of the fabric in this study. The structures shown in Figure 4 were the most successful, with the green color representing the back loops and the red color denoting the face loops.

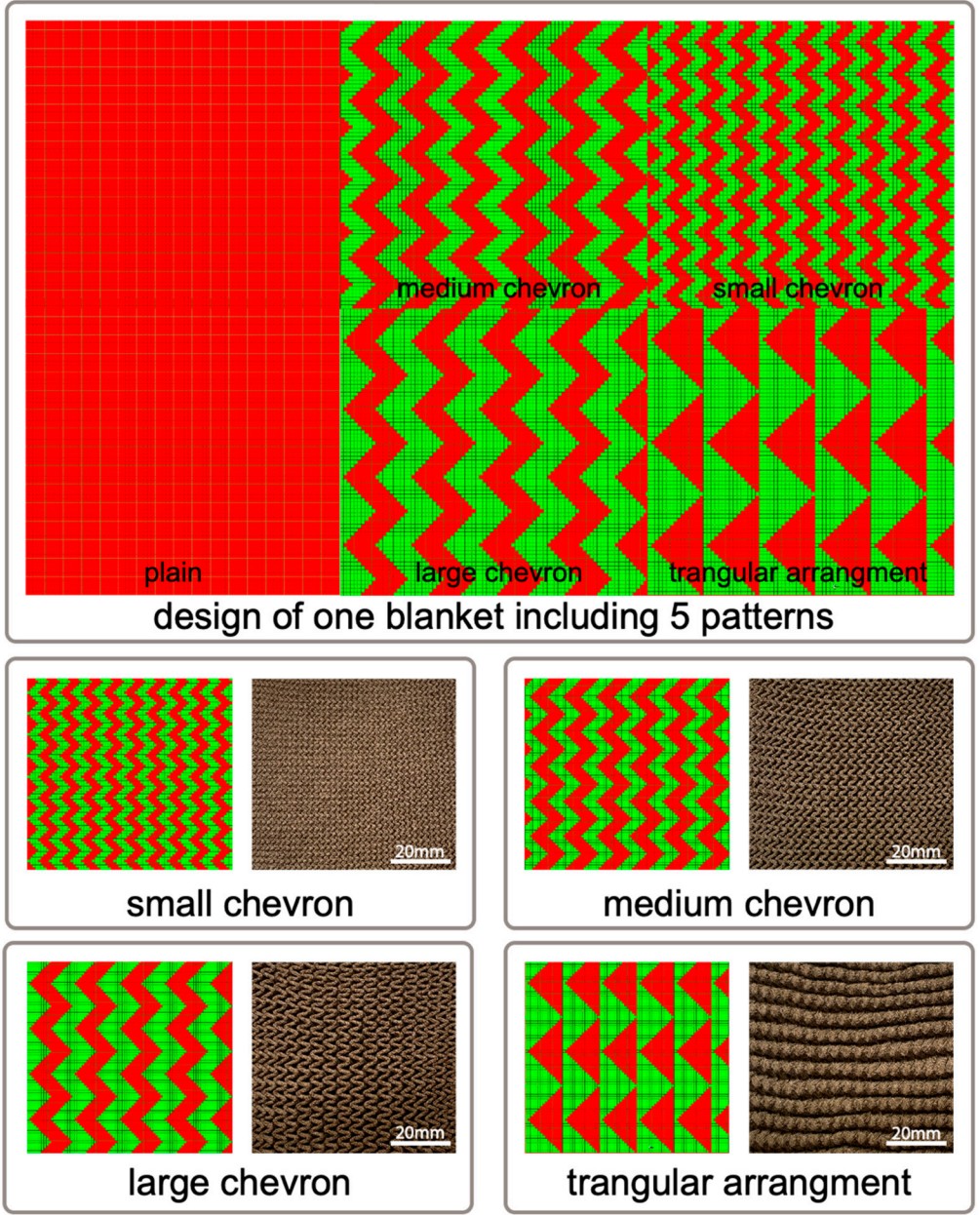

**Figure 4.** Design of the blanket including five patterns and the produced auxetic fabrics (in the last four boxes on the left, the red color is the front loop, and the green color is the back loop; the produced fabric samples are on the right).

The machine used to knit the fabric was the 14-gauge Shima Seiki N.SVR 123SP (SHIMA SEIKI MFG., Ltd., Sakata Wakayama, Japan). The yarn used included 100% 26/1 Ne Nomex®, and a 30/1 Ne Modacrylic® blend composed of 50% Modacrylic®, 30% Lyocell® A100, and 20% Twaron®. The fabrics were single jersey.

The fabric length (course direction), width (wale direction), and thickness were measured to calculate the dimensional change in the fabric to obtain the Poisson's ratio, $K_1$, and $K_2$. The average value and standard deviation (SD) were calculated. The fabric length and width were measured three times for each fabric sample. The fabric thickness was measured 10 times for each sample using a Thwing-Albert ProGage Thickness Tester (Thwing-Albert ProGage instrument company, West Berlin, NJ, USA). The method used is described in the ASTM D1777.

*2.4. Validation*

A unit strain method [9], one of the conventional methods used for measuring Poisson's ratio, was used to validate the proposed criteria. A Nikon SMZ-1000 Zoom Stereo Microscope (Nikon Metrology, Inc., Brighton, MI, USA) was used to capture the fabric images in the Phenom SEM and Forensic Textile Microscopy Laboratory at NC State University. The camera maintained the same magnification (×2) for each sample in the study. The obtained images retained the same pixel density to calculate the deformation of the fabrics, which was crucial for accuracy.

OABC is a unit area of medium chevron, as shown in Figure 5A. When the fabric was stretched in the knitting direction (OA direction), OABC will expand to O′A′B′C′. The lengths of OA, OB, O′A′, and O′B′ are recorded as $L_0$, $W_0$, $L_1$, and $W_1$ in Equation (4). Thus, the NPR is calculated using those values, which can be compared with the NPRs obtained from the proposed criterion.

## A unstreched fabric

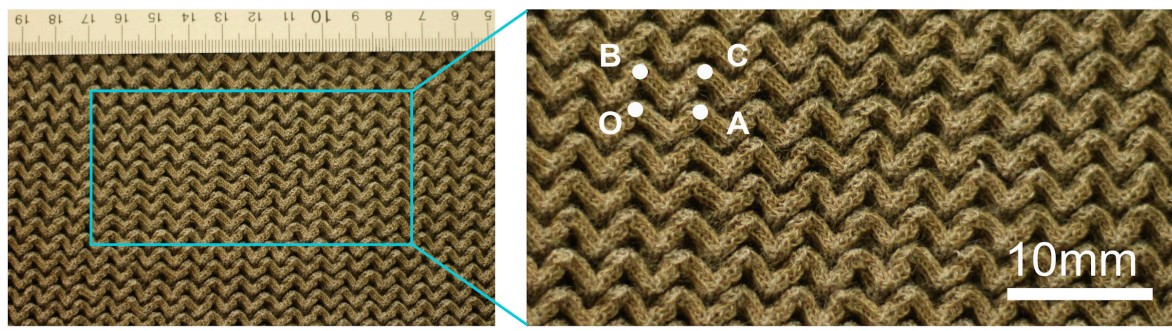

## B fabric streched in knitting direction

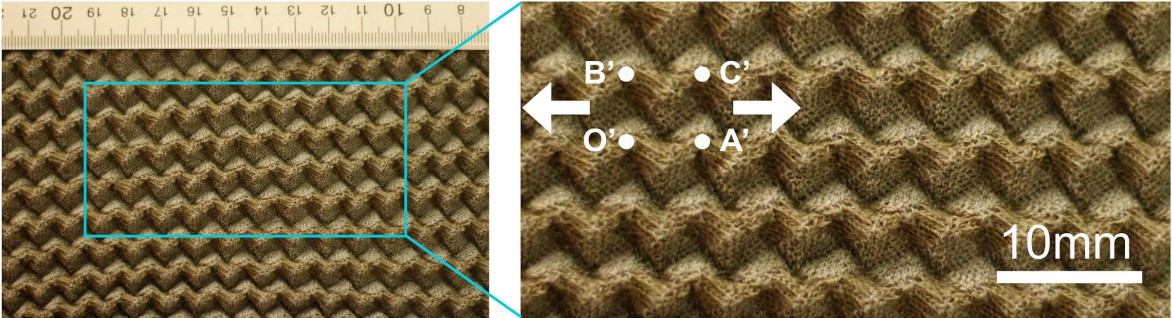

**Figure 5.** Unit strain method: (**A**) unstretched auxetic fabric (OABC is a unit area of medium chevron; O, A, B, and C are the vertices of a unit cell of medium chevron); (**B**) auxetic fabric stretched in the knitting direction (O′A′B′C′ is an expanded unit area of medium chevron after tension in the knitting direction; O′, A′, B′, and C′ are the vertices of an expanded unit cell of medium chevron).

The method was applied on an MTS Q-Test mechanical testing machine. The clamps held two edges of the fabric sample in wale directions. The speed of the top clamp was 5 mm/min, which is a quasi-static tension. The clamp movement was stopped when the fabric was stretched to a flat shape. The distances of OB, OA, BC, AC, O′B′, O′A′, B′C′, and A′C′ were measured to obtain the Poisson's ratio ($v\prime$). Each distance was measured three times, and the average value and standard deviation of the Poisson's ratio were obtained, as shown in Table A2.

The mean values and standard deviations of the conventional method and the proposed method are listed in Tables A1 and A2. Furthermore, the thickness (10 times defined by ASTM D1777), each parameter, and the Poisson's ratio ($v'$) were measured three times. The difference in parameters in Table 1 were calculated from a robust statistical analysis of

the mean of the parameters in Table A1, and therefore no mean value or standard deviation was presented.

**Table 1.** Parameters used in Equations (4)–(6) for calculating the Poisson's ratio of the auxetic fabrics ($v$) and the parameters related to fabric directionality ($K$).

|  | Single Jersey | Small Chevron | Medium Chevron | Large Chevron | Triangular |
|---|---|---|---|---|---|
| $L_1 - L_0$ | 0 | –8.07 | –8.60 | –9.73 | –7.73 |
| $W_1 - W_0$ | 0 | –10.33 | –13.30 | –13.93 | –12.90 |
| $T_1 - T_0$ | 0 | 2.64 | 3.33 | 3.19 | 3.93 |

## 3. Results

### 3.1. Dimensional Parameters

Figure 6 shows the dimensional change parameters measured from the single jersey and auxetic fabrics following doffing. Compared with the single jersey fabric, it was found that both the length and width of the auxetic fabric were decreased because of the inherent structural contraction generated from the auxetic design, as shown in Figure 6A,B. At the same time, the thickness increased, as shown in Figure 6C. Differences between the length, width, and thickness lead to structural shrinkage in an out-of-plane direction, providing additional elastic extensionality and leading to a negative Poisson's ratio.

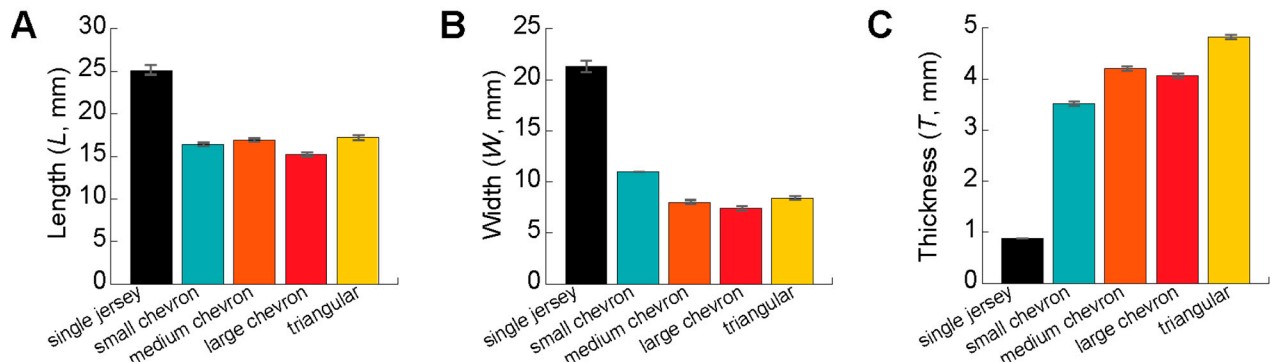

**Figure 6.** Length (*L*), width (*W*), and thickness (*T*) of the single jersey and auxetic fabrics: (**A**) length of each fabric; (**B**) width of each fabric; (**C**) thickness of each fabric (mean ± SD, see detail values in Table A1).

The relationship between the length, width, and thickness of the auxetic fabric plays a vital role in the fabric structure and auxetic behavior. The length change in the "chevrons" in Figure 6A does not show a linear change corresponding to the increasing pattern size, while the width change in Figure 6B shows a linear decrease. For the three identical patterns with different sizes, the change in thickness in Figure 6C is highly affected by the decrease in length and width in Figure 6B,C. However, the thickness change does not exhibit a linear trend corresponding to the pattern size. Among them, "medium chevron" has the highest increased thickness, indicating greater auxetic behavior. For the "triangular" pattern, although the length and width changes are smaller than those in the three "chevron" patterns, its thickness is greater.

In total, Figure 6 displays how different patterns will affect the thickness, but dimensional changes (length and width) in the in-plane direction have no proportional relationship with the increasing thickness, which is mostly due to the structural distortion of the auxetic fabric generated by the inherent yarn–yarn interactions.

### 3.2. Negative Poisson's Ratio and Auxetic Directionality

The measured dimension values in Table 1 were obtained for use in Equations (4)–(6) to calculate the Poisson's ratio and the dimensionality of the auxetic material.

Figure 7A shows the Poisson's ratio of the auxetic fabrics. All produced fabrics have Poisson's ratios smaller than −1, indicating excellent auxetic behavior in response to axial tension. Among them, "triangular" has the largest Poisson's ratio, although it has the highest thickness per unit area. This indicates that contracted structures in the in-plane direction do not fully contribute to the desired auxetic property. In addition, we proposed the directionality coefficients in Equations (5) and (6) to better understand and represent the auxetic property of the fabrics. In an ideal folding structure situation, $K_1$ could be −0.5 because the length shrinkage per unit cell would increase by half a unit of thickness. However, Figure 6B shows that the $K_1$ values do not equal −0.5, which means that the other direction contraction influences the folding structure. The $K_1$ value represents the capacity of the length direction shrinkage to contribute to the out-of-plane direction folding. The $K_2$ value represents the orientation capacity of the in-plane direction. The critical value of $K_2$ is 1. A $K_2$ value close to 1 means that the length change and width change are similar and that the fabric shrinks into a square shape.

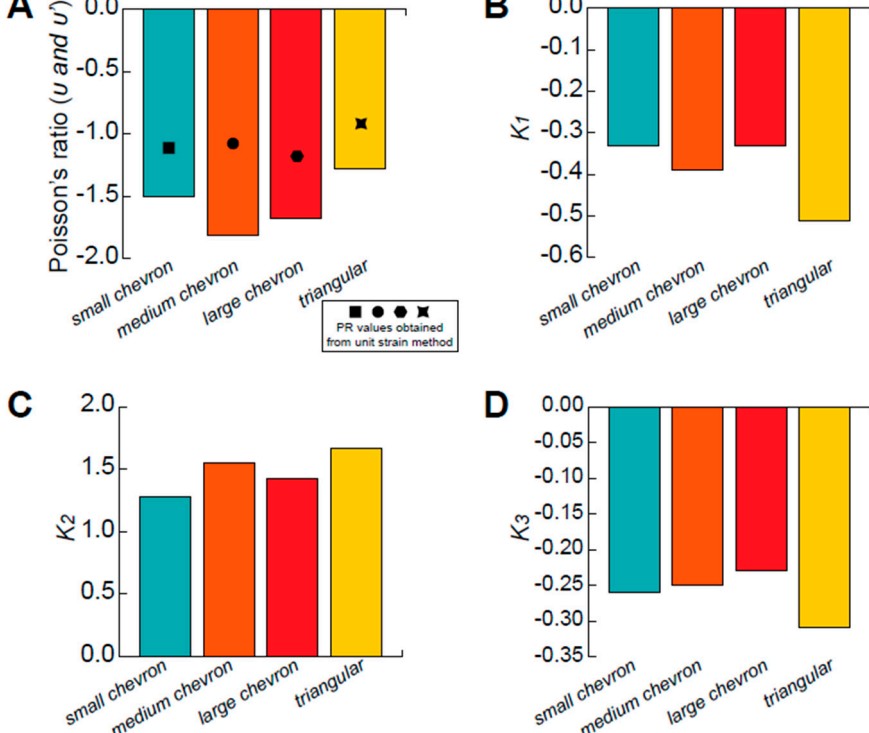

**Figure 7.** Poisson's ratio of the auxetic fabrics ($v$) and the parameters related to fabric directionality ($K$) (see the values in Table A2): (**A**) comparison between unit strain method ($v'$) and proposed criteria ($v$); (**B–D**) $K_1$, $K_2$, and $K_3$ values.

Figure 7B reveals the $K_1$ values of the auxetic fabrics. The "triangular" pattern has the lowest $K_1$ value, lower than −0.5. This indicates that both the length and width changes in the "triangular" pattern in the in-plane direction enlarge the fabric thickness. The $K_1$ value of the "chevron" pattern is lower than −0.5, which means a lower capacity to generate a contraction force that can fold the fabric into a "mountain" shape.

Figure 7C displays the $K_2$ values of the auxetic fabrics. The "small chevron" pattern has the lowest $K_2$ value, suggesting an evenly contracted structure in the in-plane direction. Figure 7D shows the $K_3$ values of the auxetic fabrics. The "triangular" pattern exhibits larger $K_2$ and lower $K_1$ and $K_3$ values, potentially contributing to its larger Poisson's ratio.

In summary, Figure 7 provides a visual representation of the Poisson's ratio and the directionality coefficients of the auxetic fabrics studied in this paper. The analysis suggests that the fabric patterns and their directionality coefficients play a significant role in determining the auxetic behavior and folding characteristics of the fabrics.

*K* values in Figure 7 were calculated from a robust statistical analysis of the mean of the parameters in Table A1, and therefore no mean value or standard deviation was presented.

### 3.3. Comparison of Unit Strain Method and Proposed Criteria

Figure 7A shows a comparison of the unit strain method (black dots) and the proposed criteria in this study. Overall, the unit strain method underestimates the NPR property of the auxetic fabrics, and there are two main reasons for this:

1. Stored elastic energy cannot be fully released to reflect the real elastic property of auxetic fabrics, which is consumed in the fiber/yarn deformations and the relative movements (slippage) among them.
2. The applied load cannot evenly propagate on the fabric due to angled fibers/yarns.

The proposed criteria inherently address the disadvantages of the conventional method. They consider the storage of the elastic strain energy inherently coming from the structural fold of the fabric. This eliminates the influence of the inaccuracies in mechanical or dimensional measurements caused by fabric deformations. The obtained NPR value is very close to the upper limit of the NPR property of auxetic fabric. Thus, it can provide an efficient estimation of the NPR property in the design and production of auxetic metamaterials.

### 4. Discussion

Auxetic metamaterials are materials that expand transversely when stretched longitudinally or contract transversely when compressed. This behavior leads to a negative Poisson's ratio, where the material expands laterally under tension. The conventional method for measuring the Poisson's ratio requires proper instrumentation, sample alignment and stabilization, and minimized boundary conditions. The method developed in this study overcomes these limitations, providing a more efficient alternative that can expedite the auxetic textile evaluation process.

Further, unlike the conventional methods that may require additional steps or equipment, the new method enables immediate analysis of the elasticity and extension properties of weft-knitted fabric. This feature provides real-time information about the fabric's mechanical response following its removal from the knitting bed. This method offers an efficient approach to measuring the Poisson's ratio of weft-knitted fabrics. The validation process and comparison with the conventional method provide evidence of the method's efficacy in accurately evaluating the Poisson's ratio for different fabric patterns.

Surface roughness is a critical property in developing biocompatible/biomedical auxetic materials. Auxetic textiles possess multiscale levels of surface roughness, such as the roughness of the fiber bundle, fabric, and 3D auxetic structure. While loaded with external loading, auxetic textiles will deform at those scale levels with respect to the different Poisson's ratios. The Kawabata Evaluation System, commonly used to evaluate fabric properties, may not accurately assess the surface roughness of thick auxetic textiles due to their complex nature. In such cases, a popular alternative is to utilize 3D image scanning techniques to capture the surface roughness in a more comprehensive and accurate manner. However, 3D scanning techniques are time-consuming and costly. The K values proposed in this paper can provide a quantitative measurement or characterization of the surface roughness of auxetic textiles.

Poisson's ratio and surface roughness are independent material properties, and both are structure-related properties. They can influence each other to some extent. Future investigation of the relationship between surface roughness and Poisson's ratio at different scale levels would provide a better understanding of the properties of auxetic textiles. That research could also help to explain the surface morphology, irregularities, and structural features of auxetic material, providing insights into its behavior and potential applications

in the biomedical field. However, it is important to note that the relationship between Poisson's ratio and surface roughness is complex and can vary depending on the specific material, loading conditions, and surface treatment. It is often studied on a case-by-case basis to understand the interplay between these properties for any given material system.

## 5. Conclusions

In this paper, we produced weft-knitted auxetic metamaterials that exhibited unique elastic deformation due to an orientated structure/pattern. We also developed a method to efficiently evaluate their NPR properties. Based on the measurement of the Poisson's ratio using the conventional method and a newly proposed method, it was found that our method is more feasible and accurate without any effects resulting from the applied load during measurement. Moreover, our method provides an evaluation of the auxetic deformation directionality, which can be used for the placement design of auxetic fabric on a garment to fulfill the requirement of deformability.

**Author Contributions:** Conceptualization, K.L. and A.W.; methodology, K.L.; validation, Z.N. and K.L.; formal analysis, K.L.; investigation, Z.N.; writing—original draft preparation, K.L. and Z.N.; writing—review and editing, Z.N., S.R., K.-L.L., and A.W.; supervision, A.W., K.-L.L., and S.R.; project administration, A.W., K.-L.L., and S.R.; funding acquisition, K.-L.L., S.R., A.W., and K.L. All authors have read and agreed to the published version of the manuscript.

**Funding:** This research was funded by the U.S. Army under the Small Business Technology Transfer (STTR) award to Advanced Cooling Technologies, Inc. (ACT), in collaboration with North Carolina State University, via contract number W9110Y 21P0012. Approved for Public Release PR2023_48786.

**Data Availability Statement:** The data presented in this study are available on request from the corresponding author.

**Conflicts of Interest:** The authors declare no conflict of interest.

## Appendix A

**Table A1.** Length (*L*), width (*W*), and thickness (*T*) of the single jersey and auxetic fabrics (mean ± SD, unit = mm).

| | Single Jersey | Small Chevron | Medium Chevron | Large Chevron | Triangular |
|---|---|---|---|---|---|
| $L_0$ | 25 ± 1.32 | -- | -- | -- | -- |
| $L_1$ | -- | 16.4 ± 0.1 | 16.93 ± 0.12 | 15.27 ± 0.25 | 17.27 ± 0.75 |
| $W_0$ | 21.33 ± 1.53 | -- | -- | -- | -- |
| $W_1$ | -- | 11 ± 0 | 8.03 ± 0.25 | 7.4 ± 0.53 | 8.43 ± 0.12 |
| $T_0$ | 0.88 ± 0.02 | -- | -- | -- | -- |
| $T_1$ | -- | 3.52 ± 0.39 | 4.21 ± 0.32 | 4.07 ± 0.46 | 4.82 ± 0.47 |

**Table A2.** Poisson's ratio of auxetic fabrics (*v*) and the parameters related to fabric directionality (*K*).

| | Small Chevron | Medium Chevron | Large Chevron | Triangular |
|---|---|---|---|---|
| *v* (Proposed method) | −1.5 | −1.81 | −1.68 | −1.28 |
| $K_1$ | −0.33 | −0.39 | −0.33 | −0.50 |
| $K_2$ | 1.28 | 1.55 | 1.43 | 1.67 |
| $K_3$ | −0.26 | −0.25 | −0.23 | −0.31 |
| $v'$ (Unit strain method) | −1.12 ± 0.35 | −1.06 ± 0.21 | −1.22 ± 0.19 | −0.92 ± 0.11 |

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
