# Peer review of "Efficient Poisson’s Ratio Evaluation of Weft-Knitted Auxetic Metamaterials"

_textiles, doi:10.3390/textiles3030018_

Round 1

Reviewer 1 Report

This is an interesting article for understanding the original concept of textile auxetic metamaterials. However it would be valuable if authors add a table representing the force and elongation at break of their of their developed auxetic textiles. It would also be good if authors discuss if Efficient Poisson’s ratio can be corelated to the surface tension and surface roughness properties of their auxetic textiles. How any change in Efficient Poisson’s ratio can affect the surface roughness of auxetic metamaterials in different levels (Macro, micro)?

Author Response

Reviewer 1

This is an interesting article for understanding the original concept of textile auxetic metamaterials. However it would be valuable if authors add a table representing the force and elongation at break of their of their developed auxetic textiles. It would also be good if authors discuss if Efficient Poisson’s ratio can be corelated to the surface tension and surface roughness properties of their auxetic textiles. How any change in Efficient Poisson’s ratio can affect the surface roughness of auxetic metamaterials in different levels (Macro, micro)?

Authors’ response:  Thank you very much for your comments. In this paper, we focused on introducing an efficient method to calculate the Poisson’s ratio of textile auxetic metamaterials, and thus few information about mechanical and surface properties of them were involved.

Structure-mechanical property relation of weft knit metamaterial was discussed in author’s previous publication (Auxetic deformation of the weft-knitted Miura-ori fold. Textile Research Journal 90 (5-6), 617-630). We used experimental data and mathematical models (geometrical model and Finite Element Analysis model) to evaluate the elastic property and of the weft knit metamaterials. Since the weft knit metamaterial is super elastic (maximum elastic strain may reach to ~200%) and the Poisson ratio is no longer negative after reaching a certain strain, we used a cyclic force-strain curve to represent the constitutive relation of their mechanical property other than a tensile curve with the breaking point. Therefore, we did not involve information about mechanical testing of auxetic materials.

Surface tension of auxetic textile is highly related to the surface roughness and yarn materials property. We have investigated the surface interaction between auxetic textile and simulated sweat liquid by using M290 Moisture Management Tester per AATCC Test Method 195. It has been found the wetting time and one-way transportation index of the auxetic textile “Large Chevron” in this paper are similar to the regular military T shirt fabric (nylon, weft knit, non-auxetic material). Due to the auxetic structure, the spreading rate of the Large Chevron is 3 times large than the regular military T shirt fabric. Thus, the Overall Moisture Management Capacity (OMMC) of Large Chevron is higher, which means a better moisture management compared to the military T shirt fabric. However, we did not correlate the surface tension to the Poisson’s ratio during the project. We will continue to work on the auxetic textile and plan an experiment to explore the relation in the future study.

The reviewer is correct. Surface roughness is a critical property in developing biocompatible/biomedical auxetic materials. Auxetic textiles possess multiscale levels of surface roughness, such like roughnesses of fiber bundle, fabric and 3D auxetic structure. While loaded with external loading, the auxetic textile will deform at those scale levels respecting different Poisson’s ratio. To authors’ knowledge, auxetic textile is too thick and surface roughness is complicated that cannot be accurately evaluated by the Kawabata Evaluation System. The popular way is to use 3D image scanning to capture surface roughness. Both surface roughness and Poisson’s ratio are structure-related property. The investigation of the relation between surface roughness and Poisson’s ratio at different scale levels would provide a better understanding about auxetic textile property. We added the discussion about this in section 4.  

Reviewer 2 Report

textiles-2320214

Title: Efficient Poisson’s ratio evaluation of weft-knit auxetic metamaterials

Kun Luan, Zoe Newman, Andre West, Kuan-Lin Lee, Srujan Rokkam

Comments:

The manuscript by West and co-workers describes a procedure for numerically evaluating the effect of elastic deformation due to orientated structure/pattern of weft knitted fabrics compared to a single jersey structure. The topic of the study is relevant in the field and its motivation is justified.

Not being an expert in this field, I cannot appreciate the practical importance of the presented results. However, several questions naturally arise.

What was the point of taking measurements on only one structural unit of the pattern? This not only requires the use of special equipment, but also increases the measurement error. On the contrary, measurements for a segment, for example, of 10 structural units will immediately include statistical averaging.

For what purpose did the authors introduce the dimensional coefficient K3, if they did not use it in any way?

Can Figure 7a and Figure 8 be combined into one figure?

Author Response

Reviewer 2

The manuscript by West and co-workers describes a procedure for numerically evaluating the effect of elastic deformation due to orientated structure/pattern of weft knitted fabrics compared to a single jersey structure. The topic of the study is relevant in the field and its motivation is justified. Not being an expert in this field, I cannot appreciate the practical importance of the presented results. However, several questions naturally arise.

Authors’ response:  Thank you very much for your comments and questions. Please see following responses to your questions.

What was the point of taking measurements on only one structural unit of the pattern? This not only requires the use of special equipment, but also increases the measurement error. On the contrary, measurements for a segment, for example, of 10 structural units will immediately include statistical averaging.

Authors’ response: In the 4th paragraph of section 2.3, we described that the fabric length and width were measured three times and the thickness was measured ten times (required by ASTM D1777) from a sample. The average values were obtained from those measurements which presented in the Figure 6. In Figure 7, the calculated Poisson ratio and K parameters are calculated from the average values. Thus, there is no statistical averaging shown in the Figure 7 results.  

For what purpose did the authors introduce the dimensional coefficient K3, if they did not use it in any way?

Authors’ response: Three K values are related to interaction of three dimensions of the auxetic fabric. K3 is similar to K1. Since it is a negative value, lower K3 represents higher shrinkage in course direction contributing to thickness of auxetic fabric. We added K3 values into Figure 7 and discuss it in text accordingly.

Can Figure 7a and Figure 8 be combined into one figure?

Authors’ response: Fig. 7a and Fig.8 has been combined into one figure.

Reviewer 3 Report

The manuscript entitled: “Efficient Poisson’s ratio evaluation of weft-knit auxetic metamaterials”, reference: textiles-2320214

General comments:

The manuscript is subject is both interesting and important. However, I read several times and still could not find clearly the explanation of the conventional method. Furthermore, I could not observe any error or deviation in both the conventional method and new proposed method. How many measurements were performed? Is the difference between them so striking? How can the authors clearly state that without a robust statistical analysis using a considerable amount of samples and measurements? Furthermore, if they are actually different why is the new method more accurate than the conventional? These are the most relevant amendments.

The manuscript does not possess line numbers which hinder the review process. There are several statements and minor issues that I would like to highlight when the manuscript possesses the line count.

Please carefully revise the entire manuscript.

Point by point suggestions:

Figure 4 first figure containing the 5 different patterns should clearly identify which of the following pictures it corresponds to. In addition, the figures should, in my opinion, contain a scale.

Figure 5 caption should include BCOA and B’C’O’A’ definitions, even though they are adequately defined in the text.

Author Response

Reviewer 3

The manuscript is subject is both interesting and important. However, I read several times and still could not find clearly the explanation of the conventional method. Furthermore, I could not observe any error or deviation in both the conventional method and new proposed method. How many measurements were performed? Is the difference between them so striking? How can the authors clearly state that without a robust statistical analysis using a considerable amount of samples and measurements? Furthermore, if they are actually different why is the new method more accurate than the conventional? These are the most relevant amendments.

Authors’ response:  Thank you very much for your comments. The conventional method used in this paper is a unit strain method showing in section 2.4 and Figure 5. The detailed measurement of Poisson’s ratio was introduced in the reference (Development of auxetic fabrics using flat knitting technology. Textile Research Journal 2011, 81, 1493-1502).  We used the same conventional method to measure the unit strain. We add more description in the section 2.4.

The mean values and standard deviations of conventional method and proposed method are listed in the Table A1 and A2. Besides thickness (10 times defined by ASTM D1777), every parameter and Poisson’s ratio are measured for three times. The difference of parameters in Table 1 and K values in Figure 7 are calculated from robust statistical analysis of the mean of parameters in Table A1 and therefore no mean value and standard deviation were presented. We added this to section 2.4. 

As the discussion in section 3.3, the reasons of the new method more accurate than the conventional are “The proposed criteria inherently address the disadvantages of conventional method. It considers the storage of elastic strain energy inherently coming from fabric structural fold. It eliminates the influence of inaccuracies during mechanical or dimensional measurements caused by fabric deformations. The obtained NPR value is very close to the upper limit of NPR property of an auxetic fabric.”

The manuscript does not possess line numbers which hinder the review process. There are several statements and minor issues that I would like to highlight when the manuscript possesses the line count. Please carefully revise the entire manuscript.

Authors’ response:  We prepared the manuscript based on the journal guidelines and template. We have revised the entire manuscript based on your comment and used the English editing service from the journal.

Point by point suggestions:

Figure 4 first figure containing the 5 different patterns should clearly identify which of the following pictures it corresponds to. In addition, the figures should, in my opinion, contain a scale.

Authors’ response: Figure 4 has been updated based on the comments.

Figure 5 caption should include BCOA and B’C’O’A’ definitions, even though they are adequately defined in the text.

Authors’ response:  Figure 5’s caption has been updated.

Round 2

Reviewer 1 Report

It is acceptable now.

Reviewer 2 Report

I have no more comments that would necessitate another review cycle.